# Modelling of ships as a source of underwater noise

Jukka-Pekka Jalkanen[1], Lasse Johansson[1], Mattias Liefvendahl[2,3], Rickard Bensow[2], Peter Sigray[3], Martin Östberg[3], Ilkka Karasalo[3], Mathias Andersson[3], Heikki Peltonen[4], Jukka Pajala[4]

[1]Atmospheric Composition Research, Finnish Meteorological Institute, 00560 Helsinki, Finland
[2]Mechanics and Maritime Sciences, Chalmers University of Technology, 41296 Gothenburg, Sweden
[3]Underwater Technology, Defence and Security, Systems and Technology, Swedish Defense Research Agency, 16490 Stockholm, Sweden
[4]Marine Research Centre, Finnish Environment Institute, 00790 Helsinki, Finland

*Correspondence to*: Jukka-Pekka Jalkanen (jukka-pekka.jalkanen@fmi.fi)

**Abstract**

In this paper, a methodology is presented for modelling underwater noise emissions from ships based on realistic vessel activity in the Baltic Sea area. This paper combines the Wittekind noise source model with the Ship Traffic Emission Assessment Model (STEAM) in order to produce regular updates for underwater noise from ships. This approach allows the construction of noise source maps, but requires parameters which are not commonly available from commercial ship technical databases. For this reason, alternative methods to fill in the required information were necessary. Most of the parameters needed contain information which are available during the STEAM model runs, but features describing propeller cavitation are not easily recovered for the world fleet. Baltic Sea ship activity data was used to generate noise source maps for commercial shipping. Containerships were identified as the most significant source of underwater noise, with a significant potential for increasing contribution to future noise emissions.

## 1. Introduction

It is recognized that anthropogenic noise might have adverse effects on the marine environment. Scientific results unequivocally suggest that animals react to sound and sometimes with devastating results (Rolland et al., 2012; Yang et al., 2008), but more commonly give rise to strong avoidance reactions (Moore et al., 2012). Not all marine life is sensitive to the same kind of noise; low frequency shipping noise (<1000 Hz) may be relevant for several fish species, whereas this range may be less relevant for marine mammals which can hear sounds up to 200 kilohertz (Nedwell et al., 2004). The problem of underwater noise was recognized by the European Commission (EC), which included sound as the eleventh descriptor in the Marine Strategy Framework Directive (MSFD) and made it analogous to pollution (European Parliament and Council of the European Union, 2008). Global maps of shipping activity help to understand that the omnipresence of waterborne traffic will

contribute to noise levels of all sea marine areas. The levels of underwater sound have been increasing since the advent of steam-driven ships (Hildebrand, 2004, 2009), but shipping is only one source of underwater noise and both natural and anthropogenic sources contribute to noise levels.

The primary source of underwater noise from ships is the propeller cavitation. Cavitation occurs when a fast rotating propeller pushes water with its blades and low pressure zone forms on the backside of the blade. Water boils and it forms collapsing bubbles which violently burst, emitting noise in the process. All propellers cavitate when rotated fast enough, but propeller design can affect how easily this occurs. The downside is that efficient propulsion and suppression of cavitation are two conflicting requirements. Currently there exists design rules (IMO, 2014) for energy efficiency of new ships, but no binding regulation to mitigate underwater noise from ships (IMO, 2012). With this setup, it is easy to understand that designing an efficient propeller is more important than designing a silent propeller, unless low noise signature is required on the battlefield (warships), or not to disturb test subjects (research vessels) (Leaper et al., 2014).

Modelling underwater noise from ships has been done for a long time and various models have been designed to describe noise sources based on measurements done since the World War II. However, these models often rely on confidential data sets, which are not necessarily available for civilian research efforts, but during the last two decades significant effort has been made to generate an experimental basis for noise model development  (Arveson and Vendittis, 2000; Kipple, 2002; McKenna et al., 2012; Wales and Heitmeyer, 2002). These data have been used to construct noise source models, which rely on parametric description of ensemble source spectra for merchant vessels. Recently, Wittekind (2014) described the noise sources using a method which describes ships as individual sources of noise which arise from individual technical features and vessel operation.

Automatic Identification System (AIS) data have been used to track exhaust emissions from ship traffic, but its use in underwater noise source modelling has been a subject of few studies where it has mostly been used to locate the noise sources relative to hydrophone setups (Hatch et al., 2008; McKenna et al., 2012). Our study carries this idea forward and builds on the development of the Ship Traffic Emission Assessment Model (STEAM) of Jalkanen et al (Jalkanen et al., 2009, 2012, Johansson et al., 2013, 2017). This approach combines the vessel level technical description, an existing noise source model (Wittekind, 2014) and ship activity obtained from AIS data, as well as facilitates the regular updates of noise source maps of any level, ranging from local to global, depending on the availability of AIS data. These data could be used to assess shipping noise, further the understanding of noise as an environmental stressor and provide tools for future sustainable governance of the sea areas.

The aim of this paper is to a) introduce a methodology for noise source mapping, which could be used for routine annual reporting of underwater noise emissions, b) provide insight on the geographical distribution of vessel noise in the Baltic Sea area and c) provide a summary of results for noise emissions from Baltic Sea shipping during year 2015.

## 2. Materials and methods

### 2.1. Ship Traffic Emission Assessment Model

The Ship Traffic Emission Assessment Model (STEAM) of Jalkanen et al (Jalkanen et al., 2009, 2012, Johansson et al., 2013, 2017) was used in this study. The Wittekind noise source model (Wittekind, 2014) was built into STEAM which facilitated noise source description based on technical characteristics of individual vessels. The selection of noise model for implementation was based on the performance of the model, availability of technical data required for proper implementation and separate description of high and low frequency contributions to source levels. Also, the Wittekind model is based on measurements which were made for a modern vessel fleet. Conceptual modelling using AIS to describe vessel activity and technical data to describe the features of vessels is independent of the choice of the source model.

The activity data used for this study consisted of 500 million AIS position reports sent by the ships sailing the Baltic Sea during year 2015. The data were provided by the member states of the Helsinki Commission (HELCOM). STEAM uses AIS to describe vessel location, time, identity and speed over ground and combines it with vessel technical data of IHS Fairplay (IHS_Global, 2016) and publicly available shipping data sources (classification societies, engine manufacturers). This combination allows for predictions of instantaneous engine power, fuel consumption and emissions as a function of vessel speed, further details of the model can be found in a recent paper of Johansson et al (2017).

### 2.2. Wittekind noise source model

Wittekind noise source model describes the ship noise as a combination of three contributions, which arise from low and high frequency cavitation and machinery noise. These are linked to vessel properties, like displacement, hull shape and machinery specifications, which is in contrast with some previously introduced ship noise models (McKenna et al., 2012; Wales and Heitmeyer, 2002). The cavitation contributions are dependent on vessel speed whereas the machinery part is not. This has important implications in noise source map generation and the time integration part of this work, which will be described in Section 2.6. The three components are described by Wittekind as

$$SL(f_k) = 10log_{10}\left(10^{SL1(f_k)/10} + 10^{SL2(f_k)/10} + 10^{SL3(f_k)/10}\right) \tag{1}$$

In (1) $f_k$ is the centre frequency of the $k^{th}$ frequency band. The SL1 (Eq (2)) represents the low-frequency cavitation noise, the second contribution (SL2; (3)) describes the high frequency cavitation and the third (SL3; Eq (4)) represents the machinery noise. In the Wittekind model, the low frequency cavitation (SL1) was obtained from fitting to experimental data (Arveson and Vendittis, 2000) :

$$SL1(f_k) = \sum_{n=0}^{5} c_n f^n + 80log_{10}\left(\frac{4c_B V}{V_{cis}}\right) + \frac{20}{3}log_{10}\frac{\nabla}{\nabla_{Ref}} \tag{2}$$

$$SL2(f_k) = -5\,ln\,f - \frac{1000}{f} + 10 + \frac{20}{3}\,log_{10}\frac{\nabla}{\nabla_{Ref}} + 60log_{10}\frac{1000c_B V}{V_{cis}} \qquad (3)$$

$$SL3(f_k) = 10^{-7}f - 0.01f + 140 + 15log_{10}m + 10log_{10}n + E \qquad (4)$$

In Eq (2), the f denotes the centre frequency of the $k^{th}$ octave band, the other constants are $c_0$=125, $c_1$=0.35, $c_2$=-8E-3, $c_3$=6E-5, $c_4$=-2E-7, $c_5$=2.2E-10 and $C_B$ denote the Block coefficient (hull form fullness when compared to a rectangular box of same length, width and depth as the ship), V indicates the instantaneous vessel speed obtained from AIS, $V_c$ represents the cavitation inception speed, $\nabla$ is the vessel displacement and $\nabla_{Ref}$ the reference vessel displacement, which is 10 000 tonnes. In Eq (4), parameters m and n represent the mass (in tonnes) and number of operating main engines, whereas E is the engine mounting parameter which indicates whether the engine is resiliently (E=0) or rigidly (E=15) mounted.

As can be seen, the Wittekind model uses parameters which are ship specific and will lead to individual noise source description depending on vessel features, but some of these are not available from ship databases which provide other vessel specifications. However, there are numerous parameters which need to be derived during the noise source calculations. Some of these, like $c_B$, $\nabla$ and n, are already calculated during a regular STEAM run, but engine mass (m), mounting parameter (E) and cavitation inception speed ($V_{CIS}$) were determined as described in Sections 2.3, 2.4 and 2.5.

### 2.3. Main engine mass

Main engine mass is not routinely included in commercial ship databases and we have augmented the STEAM database with engine masses obtained from technical documentation of engine manufacturers and engine catalogues (Barnes et al., 2005). Engine mass could be determined explicitly for about two thirds of the global fleet. For the remaining cases, a linear function was developed to estimate engine mass based on the size (installed power) of engines. For four-stroke engines, main engine mass is determined by multiplying the installed kW/engine with 0.0155 which corresponds to 65 kW/ton power/mass ratio and falls between the range of values proposed by Watson (Watson, 1998). Cylinder arrangement (in-line vs V arrangement) has an impact on predicted mass, because in-line engines tend to be heavier than V engines which leads to lower power/mass correlation than in case of two stroke engines. This does not apply to 2-stroke engines, because only in-line engines are used.

There are about 19 600 vessels equipped with four stroke engines, mass of which needs to be evaluated with the proposed power/mass methodology. The quality of linear fit is slightly worse for four stroke engines ($R^2$=0.814) than for two stroke engines ($R^2$=0.955) because of variable cylinder arrangement described above. There are 24 300 vessels with two-stroke engines the mass of which can be determined from manufacturer documentation. Mass of two-stroke main engines for 5500 ships need to be estimated based on installed engine power (in kW). Further, there are 3100 vessels for which engine stroke

type is unknown. In unknown cases, most similar vessel details (Johansson et al., 2017) are used to determine missing technical data.

For two-stroke engines, engine power output is multiplied with 0.0322 (red line). For example, Man B&W 10K98MC-C engine, the predicted mass is 1725-1797 tonnes, whereas manufacturer specifications indicate mass of 1854 tonnes. Watson recommends 0.035-0.045 tonnes/kW (upper and lower black lines of Figure 2). It should be noted, that the range recommended by Watson (1998) leads to higher engine masses than the best fit to engine setup of the current fleet of 24 300 samples.

For gas turbine machinery, 0.001 tonnes/kW should be used according to Watson. There are 480 entries in the ship database, which indicate the use of turbine machinery, either gas or steam versions. The accuracy of mass predictions for vessels equipped with turbine machinery is poor. No correlation was found between engine mass and power output. The Watson recommendation was adopted and 0.001 tonnes/kW was used for all turbine machinery. It should be noted that the applicability of the Wittekind noise source model to turbine machinery is an extrapolation of original results and are likely to result in large uncertainties.

### 2.4. Engine mounting

Unfortunately, engine mounting parameter is not available in the available technical databases. Main engines of a ship can be bolted directly to the rigid box girder without additional damping material to absorb vibrations of engines. This is known as rigid mounting and it is usually applied to large two-stroke engines but can also apply to some large four-stroke engines. Resilient mounting of the engine is used if it is necessary to reduce structure-borne vibrations or noise which would otherwise be transmitted to the hull. According to Rowen (2003) and Kuiken (2008), resilient mounting is usually applied to medium and high speed diesels, which are sufficiently rigid in bending and torsion. In this work, all two stroke engines have been assigned "rigid mounting" status and "resilient mounting" is assumed for all four stroke engines, although some of the four-stroke engines can be installed both ways (Wartsila, 2012, 2015 2016). We investigated the impact of these assignments on emitted noise levels to several kinds of ships. Source level curves for some of these cases can be found in Supplementary Material.

### 2.5. Cavitation inception speed

The description of cavitation is, among other factors, a function of propeller disc area and propeller tip speed. Commercial ship databases do not contain enough information, like the number of blades and diameter, of propellers installed to ships to generate the cavitation inception speed. An alternative method to determine this parameter was developed based on discussions with a manufacturer of propulsion equipment. Based on these discussions, an approach based on vessel Block coefficient and design speed was developed (5).

$$V_{CIS} = min\{max[(1.42 - 1.2c_B) * V_d; 9]; 14\} \tag{5}$$

where $V_{CIS}$ is the cavitation inception speed (knots), $C_B$ is the Block coefficient and $V_d$ the design speed of the vessel (knots). Between 9 and 14 knots the cavitation inception speed is a linear function of the Block coefficient (hull shape). According to Eq (5), all ships will cavitate at 14 knots; especially the fast RoPax, cruise ships and most modern containerships will fall into

5    this end of the range. This contrasts with most bulk cargo carriers and tankers for which $V_{CIS}$ is close to nine knots. With these extremes, there are various exceptions, for example very large containerships (over 18 000 TEU capacity) and new LNG carriers which perform less well and have lower inception speed than most of the ships of their type.  It is unclear why this occurs, but there is a known trade-off between propeller efficiency and noise (Carlton, 2010). The gradually tightening energy efficiency requirements for ships may lead to ships which are noisier than their predecessors if low noise emissions are not

10   considered as a meaningful parameter during the design phase. Highly efficient propellers may not be the most silent ones.

### 2.6.  Noise source map generation

To represent underwater noise emissions as a map, an approach was developed to facilitate this form of emission reporting. The source level is related to the power emitted ($P_k$) in frequency band k, as:

$$SL_k[dB\ re\ 1\ m, 1\ \mu Pa] = 10log_{10}\frac{P_k}{P_{Ref}} \tag{6}$$

where $P_{Ref} = \frac{4\pi p_{ref}^2}{\rho c}$ is a reference power, $\rho$ and c are density and speed of sound while $p_{ref}$=1 µPa. Assuming that all noise sources are uncorrelated, the total emitted power from all M ships in area A at time t is given as:

$$P_k^{tot}(t) = \sum_{m=1}^{M} P_{k,m}(t) \tag{7}$$

where $P_{k,m}(t)$ is the sound power (in J s$^{-1}$) emitted by ship m. This quantity is additive and facilitates the summation of ship specific noise energy over a specific period (in Joules). The sound power map is more of a visual aid than a direct input dataset for noise propagation modelling, which usually demands point source descriptions of the noise sources. For examples of propagation modelling from multiple ships, facilitating the evaluation of the sound pressure level in arbitrary point in the water

25   column, the reader is referred to e.g. Karasalo et al. (2017) and Gaggero et al. (2015). Presenting sound energy as geographically distributed quantity will help visualizing noisy areas, as has also been investigated by Audoly (2015). Similar

to the emission maps of atmospheric pollutants, noise source maps should not be taken as a representative description of underwater noise any more than an emission map of $NO_x$ does not describe airborne pollutant concentrations. The maps presented in this work describe the noise sources, not underwater propagation of noise. It should be noted that the numbers presented as a map are a function of grid cell area and should be normalised to unit area. In this work we have used one square kilometre as grid cell size.

Ships spend a significant part of their activity in harbour areas (Smith et al., 2014). The time integration step (Eq (7)) leads to a situation where harbour areas were represented as significant sources of underwater noise. This is a feature of the machinery contribution of noise source description (see Eq (4)) which remains non-zero when ships are standing still. With the current approach it is not possible to distinguish between ships standing still with engines on or off. The Wittekind noise source model is intended for moving vessels and application of this model to stationary vessels would have been a clear extrapolation of the original intention. For that reason, we chose to only apply the time integration of sound power for moving ships. In STEAM, time integration of sound power is applied only for cruising and manoeuvring modes of vessel operation and stationary vessels do not contribute to total sound energy regardless of the fact that there may be auxiliary engines running during harbour visits which may contribute to the emitted underwater noise. Noise from auxiliary engines is not modelled in this approach even if they may be a significant source of atmospheric noise in harbour areas. With these definitions, a source emitting one megajoule of noise in one year corresponds to a continuous monopole source with approximately 156 dB re 1 µPa at 1 m sound pressure level, assuming that free-field approximation is valid.

## 3. Results and discussion

### 3.1. Shipping noise emissions in the Baltic Sea area

The noise maps were generated for 1/3 octave bands which have 63, 125 and 2000 Hz central frequencies (Van der Graaf et al, 2012). The two lowest bands are relevant to various fish species whereas the 2 kHz band is relevant for marine mammals (Nedwell et al., 2004; Nikopouloulos et al, 2016). Using the methodology described above, the generated noise source maps for Baltic Sea shipping in 2015 (for 63 Hz band) are depicted in Figure 3.

As can be seen from Figure 3 noise source maps have noise hotspots on the main shipping lane in the Danish Straits, between islands Fyn and Sjælland. Also, outside Kiel and Rostock harbours high values for sound energy were estimated. The annual noise energy emitted in the 63 Hz band was 117 gigajoules during 2015 and highest contributions were from bulk cargo and container ships as well as tankers. The noise emissions increase towards the end of 2015. Maximum monthly noise energy is emitted in December 2015, 32 GJ/month whereas the minimum occurs in February, 25 GJ/month. These are summed energies over all three bands, 63, 125 and 2000 Hz. Daily noise energy emissions of January are 0.86 GJ/day, but emissions towards the end of year 2015 already exceed 1 GJ/day (the daily maximum occurs in October, 1.07 GJ/day). These indicate 20% growth in noise energy emissions (in gigajoules, not dB) during 2015.

Plotting noise energy emitted by each ship type, relative to total noise energy emitted at each band, indicates that containership and bulk cargo carriers are the two largest sources of underwater shipping noise in the Baltic Sea area. Containerships represent about three percent of all ships, but are responsible for 27 % of the noise emitted at 125 Hz band. Bulk cargo carriers also have high share of noise emissions, but bulkers represent a larger share of the total numbers of ships

(8%). (Figure 4; Table 1). Analogous to energy efficiency metrics, reported in grams of $CO_2$ emitted per amount of cargo carried and distance travelled (in g ton$^{-1}$ km$^{-1}$), the emitted noise energy should also be compared to transport work or distance travelled. If done this way, containerships represent 15% of the transport work and emit 23% of the noise energy (sum of noise energy emitted at 63, 125 and 2000 Hz bands). In case of bulk cargo ships, the share of noise energy emissions is 23% and share of the transport work done is 21%. Considering the large share of transport work, bulk and general cargo ships emit less

noise than containerships. The largest discrepancies between noise energy emitted and distance travelled occur with RoPax vessels, which are responsible for three percent of the transport work and contribute nine percent of the noise energy (sum of energy over all three bands) emitted in the Baltic Sea area. If noise efficiency index is defined as joules of noise energy emitted for each ton km of cargo carried, noise efficiency index in mJ ton$^{-1}$ km$^{-1}$ is very high for RoPax vessels (920 millijoules ton$^{-1}$ km$^{-1}$) whereas for containerships and bulkers these are 491 and 360 mJ ton$^{-1}$ km$^{-1}$, respectively. With this metrics, lowest index

is achieved with slow moving vessels, like general cargo carriers and crude oil tankers, which emit less than 200 millijoules of noise energy per ton km carried.

For most cargo ships $V_{CIS}$ is predicted to be close to nine knots, except for containerships, and about one quarter of these slow vessels sailed in 2015 slower than their predicted cavitation inception speed (Figure 5). If the cargo carrying fleet in the Baltic Sea area returns to normal operation with speeds closer to their design speed, it is very likely that a significant increase

in noise energy will be seen for the quarter of the cargo fleet now operating at slower speeds than their $V_{CIS}$. This increase could happen without increasing the fleet size at all. A significant portion of oil product tankers and cruise vessels were operating with speeds lower than their cavitation inception speed. It may very well be that the contribution from oil tanker fleet may increase when the slow operating vessels speed up again, but their overall contribution to sound power is quite low, only about two percent. However, if all of the twenty percent of containerships which in 2015 operated under their $V_{CIS}$ speed

up, the impact on sound energy increase will be significant, because containership contribution to overall sound power is high. Voluntary speed reduction was also observed in the Third IMO GHG study (Smith et al., 2014), especially in the container ship class of ships. Speed reduction may occur in situations where vessels may not be fully loaded, overcapacity in the market exists and costs can be lowered by sailing slower than the design speed. The required power, and also the fuel consumption, are cubic functions of speed and speed reductions may lead to significant savings if vessel schedules allow it.


### 3.2. Uncertainty evaluation

Karasalo et al (2017) tested the performance of the Wittekind noise source model with inverse modeling from hydrophone measurements. The transmission loss of the measured noise signature was modelled using XFEM code (Karasalo, 1994) to

obtain the noise source at reference distance. In their paper, Karasalo et al (2017) observed a good fit between the Wittekind predictions and observed signals for cargo ships and tankers and tugboats, but larger differences were observed with passenger and RoRo vessels for which the Wittekind model overestimated the noise source levels. It is very likely that this is because the Wittekind model was mainly intended for large ocean-going vessels with a single fixed pitch propeller or a single controllable

pitch propeller when they are operated close to their design pitch (Wittekind D, Oct 2017, personal communication). Voluntary operation of a vessel with lower speed (slow steaming) may work as a noise mitigation option for deep ocean vessels with a single fixed pitch propeller, but it may not work with ships equipped with controllable pitch (CP) propellers and it may lead to higher than expected noise emissions (Wittekind, 2009).

Significant uncertainty may be involved in the estimation of the cavitation inception speed ($V_{CIS}$), which is not readily

available from any of the ship databases directly and was estimated using the vessel design speed and hull form (see Eq. 5). Contribution of $V_{CIS}$ to vessel noise source level is significant, because at speeds below this threshold value vessel noise is notably lower than above it. We tested the impact of $V_{CIS}$ uncertainty by testing the sensitivity of predicted noise to cavitation inception speed by altering the lower and upper bounds of Eq (5) to ten and 15 knots. This increased the speed where propellers cavitate and would lead to larger portion of the fleet operating at non-cavitating conditions than under default assumption. The

differences in predicted noise energy in the Baltic Se area were most pronounced in the low frequency band (63 Hz), where the total noise energy emitted was decreased by 26% when higher values of $V_{CIS}$ were applied. For all considered frequency bands, the total reduction was 19%. Sum of energy emitted at higher frequency bands was also decreased, by seven percent for both 125 and 2000 Hz bands, respectively. Change of cavitation speed range altered the noise energy emissions from RoPaxes only by seven percent and results for passenger cruise vessels were unchanged. This is probably because RoPax and

cruise vessels mostly operate at speeds larger than 15 knots and cavitation still occurs regardless of the higher $V_{CIS}$ tested here. For containerships, noise emissions were reduced by 19%, but largest changes (-39%) occurred in the tanker class of ships. Contributions from other slow-moving vessels, like cargo ships were also significantly reduced (-27%).

The uncertainty concerning $V_{CIS}$ can be reduced with more in-depth research on cavitation inception. Findings from such studies should be released as open access reports and datasets to facilitate further research on underwater noise emissions. In

case of controllable pitch (CP) propellers the speed of the vessel is regulated with the propeller pitch and not necessarily adjusting the rotational speed of the propeller. Without additional information about the marine propellers used in the ships it is difficult to assess the details of cavitation. Modern passenger vessels are usually equipped with multiple four-stroke engines and have more than one propeller, often CP type. In 2015, about ten percent of the vessels sailing the Baltic Sea were equipped with two or more propellers and the contribution of these ship types to the total noise energy in 125 Hz frequency band was

around 13%. It is likely that the accuracy of noise emission of the passenger vessel fleet is worse than that of the cargo ships, but this will not change the main conclusions of this paper.

The Wittekind model was built for vessels with a single propeller and a four-stroke main engine. Application of the Wittekind model to large two-stroke engines commonly propelling the global fleet, may lead to increased uncertainty in predicted source levels. Most (82%) of the commercially operated vessels in the Baltic Sea use four stroke engines and the

great majority (90%) is equipped with a single propeller. The Wittekind model does not include contributions from auxiliary engines, which may be a significant noise source in port areas. This was one of the reasons this contribution has been exempted from time integration of noise energy. Neglecting the continuous time integration during harbour visits will also produce some uncertainty to final results, but the magnitude of this contribution is difficult to estimate because the current approach will not be able to distinguish between ships anchored with their engines shut down and ships which keep their engines running even when vessels remain still. It is very likely that harbour areas are not significant fish or marine mammal habitats, which should reduce the significance of this uncertainty concerning the consequent noise impact assessments on marine life.

## 4.   Summary

Underwater noise is rarely a design parameter for new ships, unless warships or research vessels are considered, and only voluntary guidelines to mitigate vessel noise exist. Currently, for the commercial fleet, efficiency of the propeller is more important than low noise emissions and these two conflicting requirements may lead to worse noise problems when more energy efficient designs are required. Cavitation of propellers is usually avoided to alleviate mechanical problems arising from erosion, not to mitigate noise emissions.

A methodology was presented to derive underwater noise emissions from ship activity and technical data. This facilitates annual updates of noise source maps for frequency bands of 63, 125 and 2000 Hz regardless of the study scale. With global AIS data, also global noise source studies are possible.

For the Baltic Sea during 2015, the most significant noise sources are the bulk carriers and containerships. Container vessels represent about three percent of the total number of IMO registered vessels but are responsible for one quarter of noise energy emitted, which makes them the largest contributor to vessel noise in the Baltic Sea area. It was discovered that about 20% of the containerships currently operate on speeds below the estimated cavitation inception speed. If these vessels increase their operating speed closer to their design speed, a significant increase of underwater noise may occur in the Baltic Sea area without increasing the fleet size at all. However, the containership share of the total transport work is almost as large as containership noise contribution. Considering the distances travelled and cargo carried, RoPax vessels have disproportionally large contribution to vessel noise. It is unclear how well the current approach can be applied in multi-propeller, multi-engine cases for which the Wittekind noise model was not originally intended. Further work is needed to understand the performance of current noise modelling tools in these cases.

It is unclear what kind of physical impact the current level of shipping noise has on marine life in the Baltic Sea area. Shipping is only one source of underwater noise and many other sources exist, both natural and anthropogenic. Noise is not routinely monitored, but it is measured in many research projects concentrating on underwater noise. There are no long-term observations of noise which could be used to determine how noise levels have developed in the Baltic Sea in the past years, but AIS data is available for at least for the last decade. This enables noise modelling studies covering this period. In general, modelling must rely on robust experimental data, which should be available to assess the performance of the modelling work.

Currently, only limited opportunities to do this exist from a handful of research projects; national measurement networks and international cooperation are needed. The noise source emission maps are available in the SHEBA project data portal (http://sheba.hzg.de/thredds/catalog.html).

## 5   **Author contributions**

JPJ was responsible for overall coordination of the work, the Wittekind noise model adaptation for STEAM and main contribution to this paper. LJ was responsible for technical implementation of the noise module and running the STEAM model. ML and RB provided technical expertise in noise model selection and adaptation. PS, MÖ, IK, MA were responsible for developing a methodology for noise source mapping and consecutive noise propagation modelling, which contributed to 10   the uncertainty evaluation. HP and JP provided expertise on relevant impacts on marine life and contributed to noise source mapping method development.

**Acknowledgements**

This work resulted from the BONUS SHEBA project and it was supported by BONUS (Art 185), funded jointly by the EU, the Academy of Finland, Swedish Agency for Marine and Water Management, Swedish Environmental Protection Agency 15   and FORMAS. We are grateful to the HELCOM member states for allowing the use of HELCOM AIS data in this research.

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

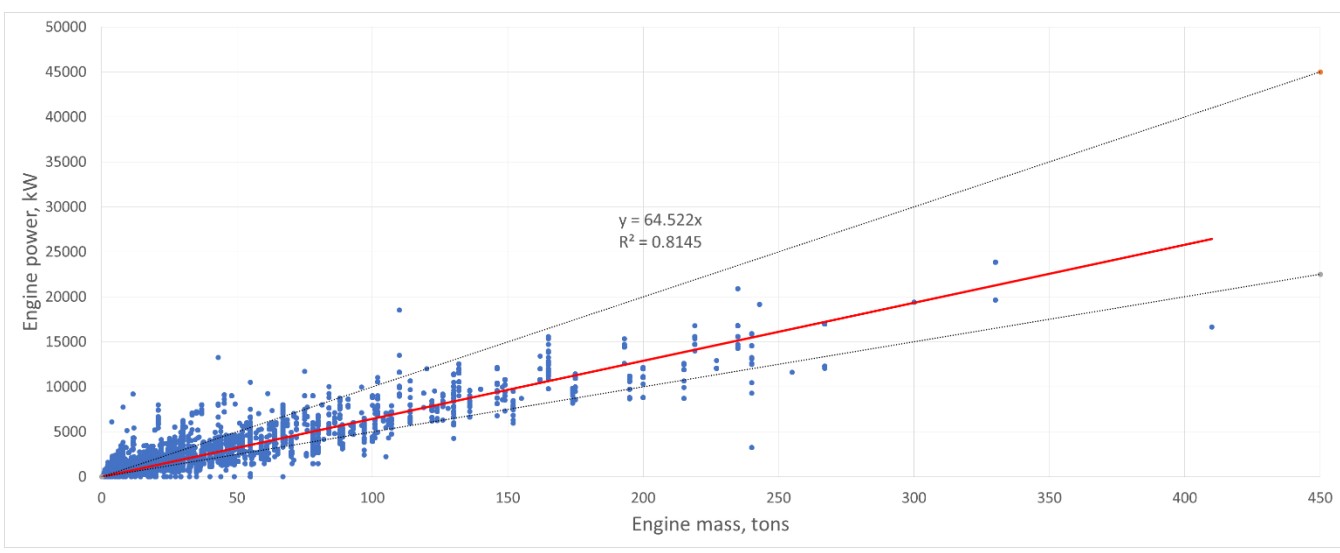

**Figure 1 Predicted and actual main engine masses of 31 500 four stroke engines. The black lines represent the range given by Watson (Watson, 1998). The red line indicates the mass/power dependency used in this study for cases where engine mass could not be determined**

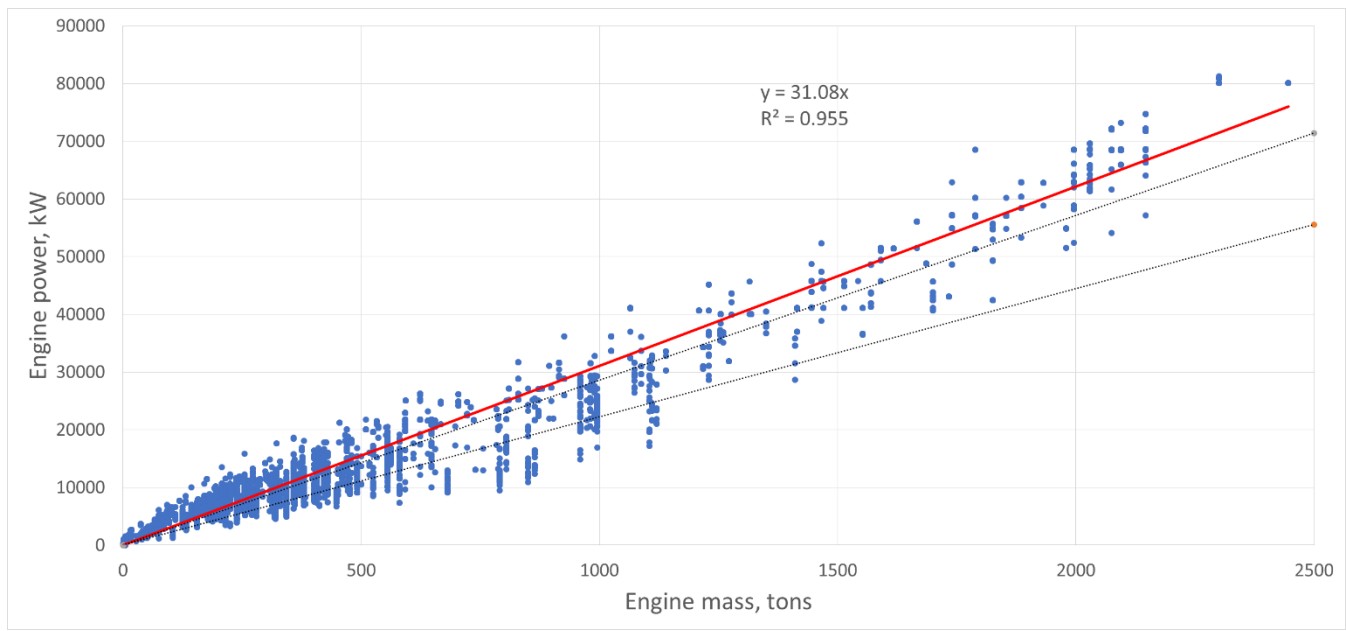

**Figure 2 Predicted and actual main engine masses of 24 000 two stroke engines. The black lines represent the range given by Watson (1998). The red line indicates the mass/power dependency used in this study.**

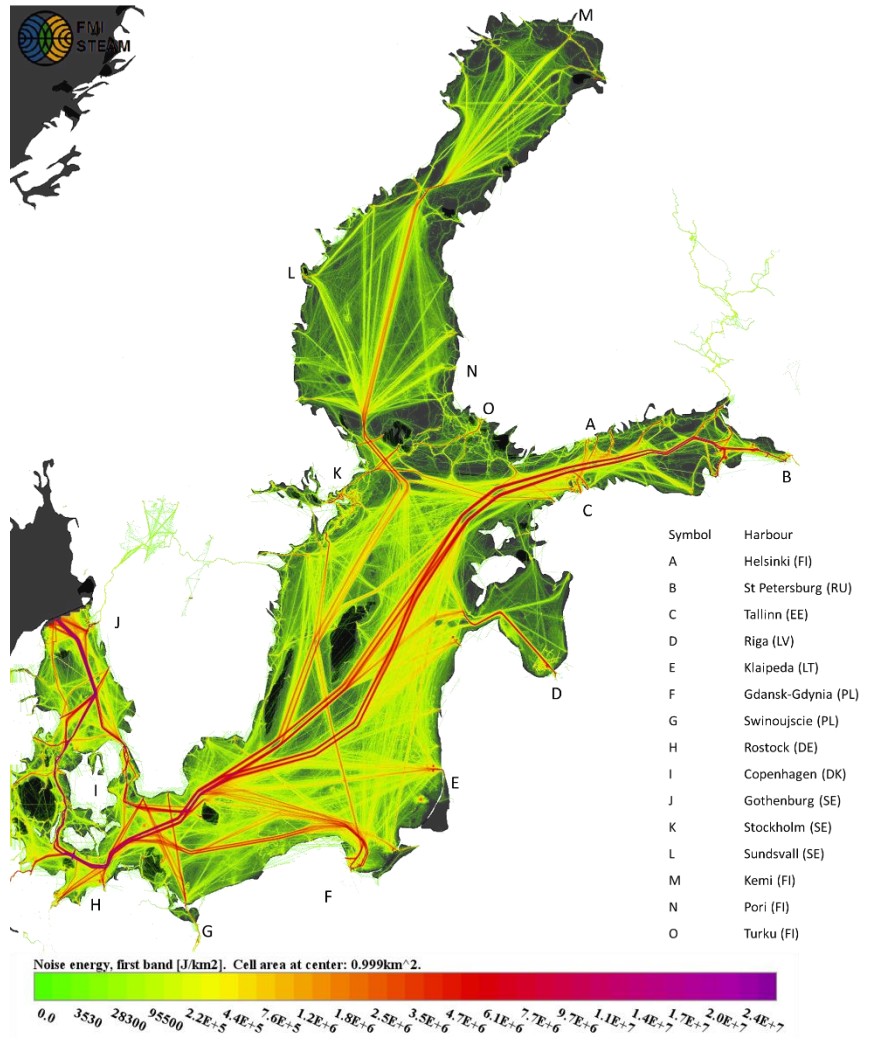

**Figure 3 Noise source map for Baltic Sea shipping. This map indicates sum of sound energy in units of Joules per grid cell (cell area 1 km$^2$) during the year 2015.**

| Symbol | Harbour |
|---|---|
| A | Helsinki (FI) |
| B | St Petersburg (RU) |
| C | Tallinn (EE) |
| D | Riga (LV) |
| E | Klaipeda (LT) |
| F | Gdansk-Gdynia (PL) |
| G | Swinoujscie (PL) |
| H | Rostock (DE) |
| I | Copenhagen (DK) |
| J | Gothenburg (SE) |
| K | Stockholm (SE) |
| L | Sundsvall (SE) |
| M | Kemi (FI) |
| N | Pori (FI) |
| O | Turku (FI) |

Noise energy, first band [J/km2]. Cell area at center: 0.999km^2.

0.0  3530  28300  95500  2.2E+5  4.4E+5  7.6E+5  1.2E+6  1.8E+6  2.5E+6  3.5E+6  4.7E+6  6.1E+6  7.7E+6  9.7E+6  1.1E+7  1.4E+7  1.7E+7  2.0E+7  2.4E+7

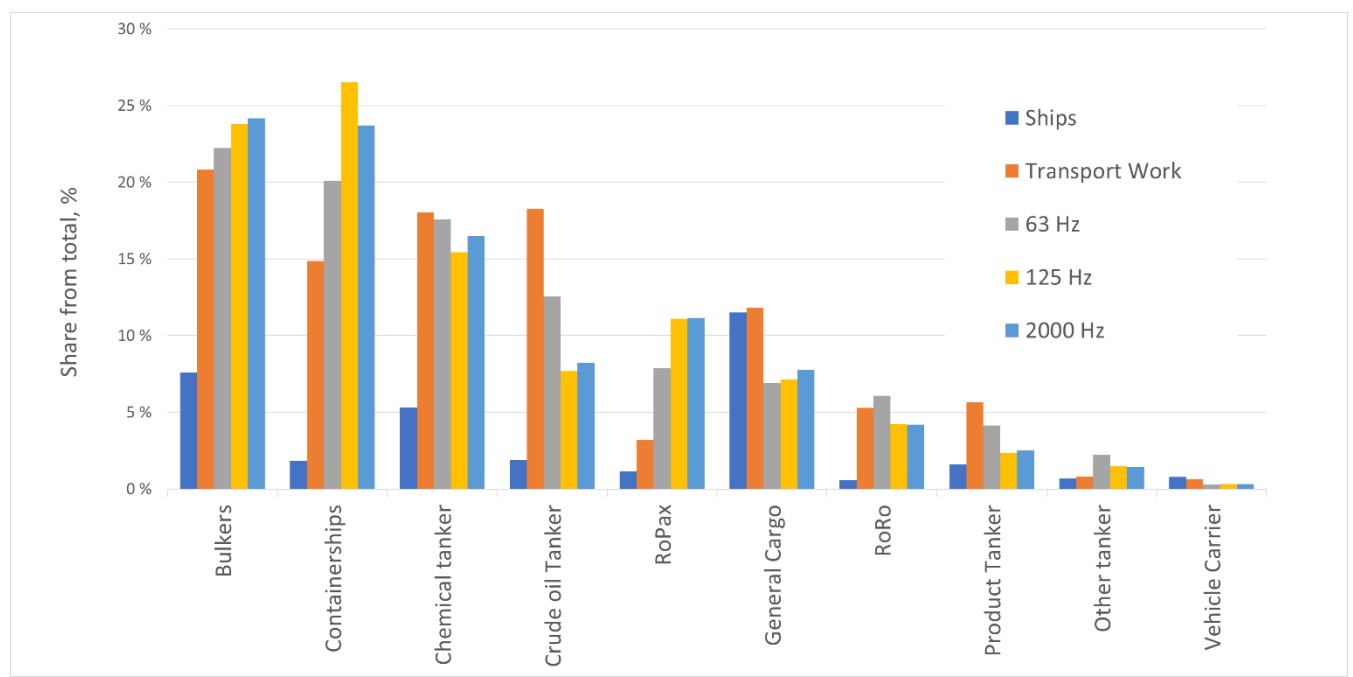

**Figure 4 Contribution of different ship types to annual emissions of underwater noise energy (share of energy emitted at 63, 125 and 2000 Hz bands). Dark blue bar=share of specific type of ships from all ships; Orange=Share of transport work; Grey, Yellow, Light Blue=Share of noise energy emitted by ships of each type from total energy.**

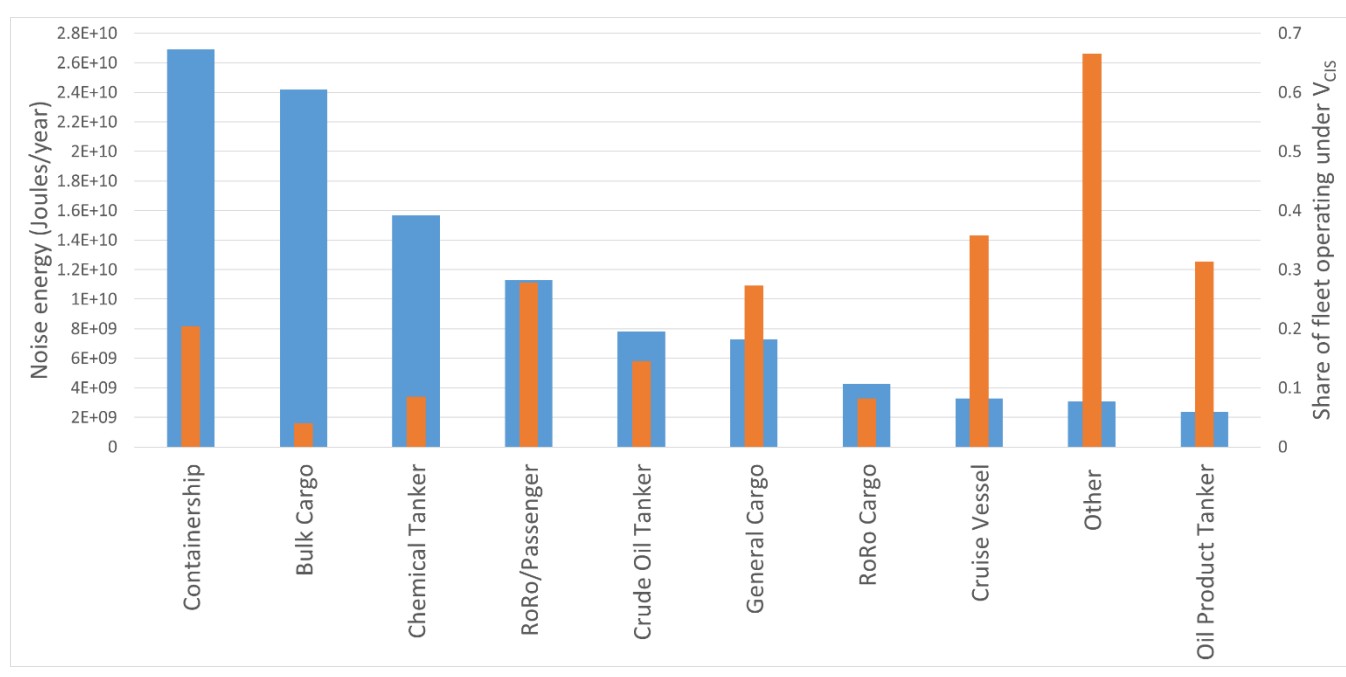

5 **Figure 5 Noise energy emitted by different ship types in 125 Hz frequency band (in Joules per year; blue bars, left axis). The share of the fleet operating under cavitation inception speed is also indicated (orange bars, right axis). For example, containerships are the biggest source of the Baltic Sea fleet with 13 gigajoules of sound power emitted. Of the containership fleet, about 20% operate with speeds lower than their predicted cavitation inception speed.**

**Table 1 Noise energy emitted by various ship types in the Baltic Sea area during the year 2015. The top ten contributors are reported, these represent over 90% of the noise energy emitted.**

| Type | Noise energy (GJ/a), 63 Hz | Noise energy (GJ/a), 125 Hz | Noise energy (GJ/a), 2 kHz |
|---|---|---|---|
| **Bulkers** | 48.4 | 24.2 | 0.4 |
| **Containerships** | 43.7 | 26.9 | 0.4 |
| **Other tanker** | 4.9 | 1.5 | 0.0 |
| **RoRo** | 13.2 | 4.3 | 0.1 |
| **RoPax** | 17.1 | 11.3 | 0.2 |
| **General Cargo** | 15.0 | 7.3 | 0.1 |
| **Vehicle Carrier** | 0.6 | 0.3 | 0.0 |
| **Product Tanker** | 9.0 | 2.4 | 0.0 |
| **Chemical tanker** | 38.3 | 15.7 | 0.3 |
| **Crude oil Tanker** | 27.3 | 7.8 | 0.1 |
| Total | 237.4 | 116.6 | 1.7 |

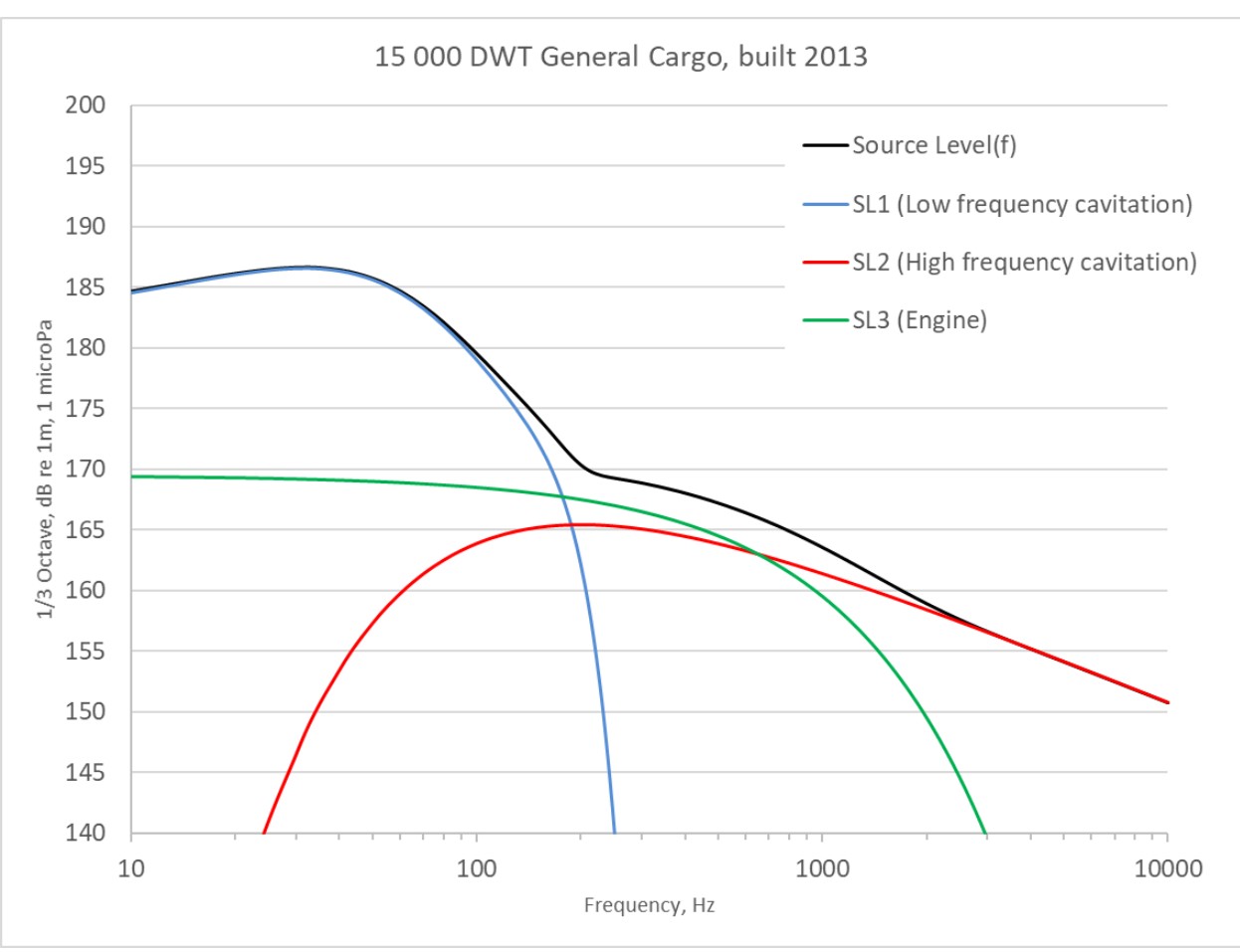

**Figure S6 Source levels for a 15 000 DWT General Cargo vessel with 4-stroke engine and a FP propeller at design speed of 14.5 knots.**

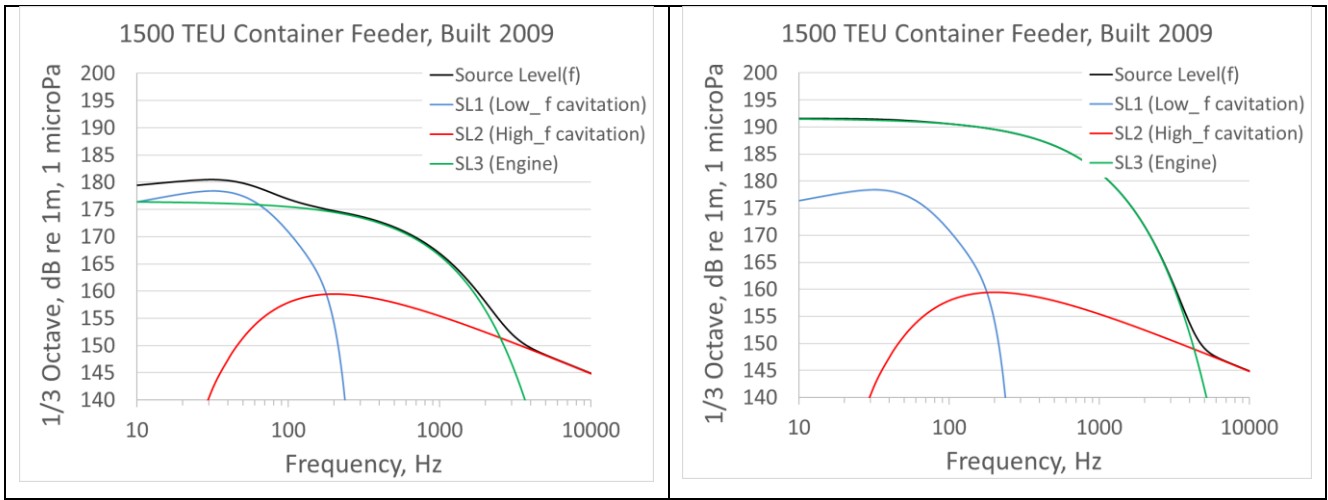

**Figure S7 a and b. Noise source levels for a 1500 TEU Container feeder vessel with a 2-stroke engine and a FP propeller assuming flexible mounting (a, left) and rigid mounting (b, right). Vessel traveling at design speed of 19.8 knots**

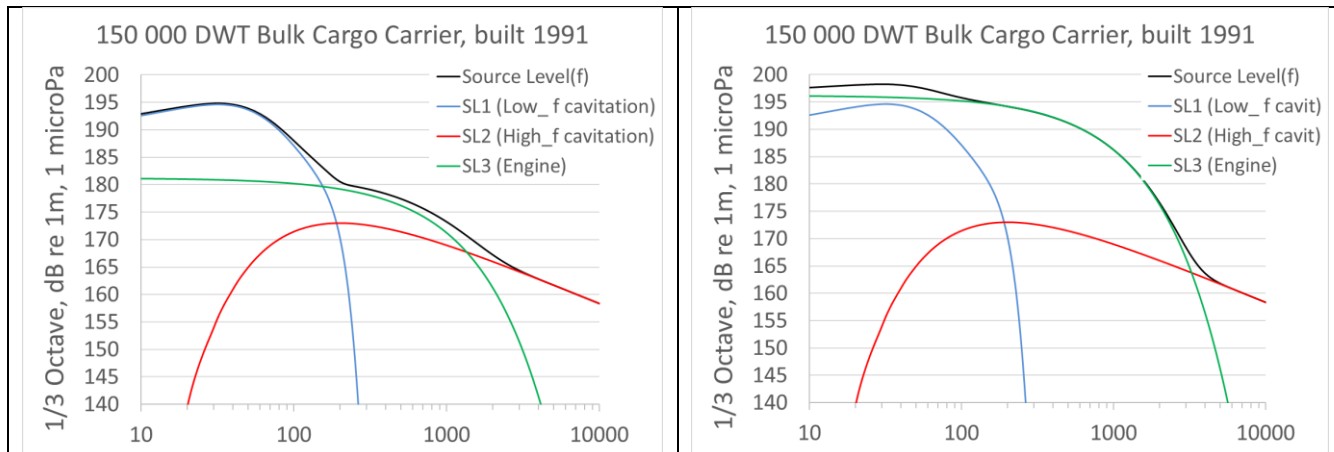

**Figure S8a and b. Noise source levels of an 150 000 DWT Bulk cargo carrier with a 2-stroke engine with a FP propeller. Source levels estimated assuming flexible mounting (a, left) and rigid mounting (b, right), with vessel traveling at design speed of 13.7 knots.**

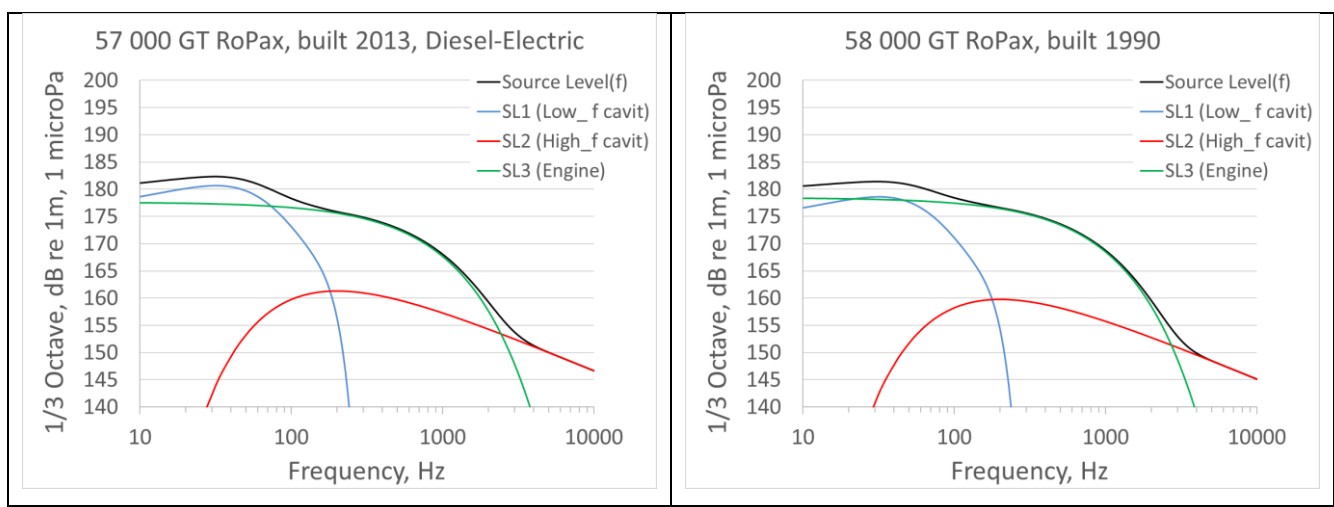

**Figures S9 and S10. Noise source levels of a 57 000 GT RoPax with four 4-stroke main engines driving two electrical motors with two CP propellers, traveling at design speed 21.8 knots (left) and 58 000 GT RoPax, which has four 4-stroke main engines and two CP propellers, at design speed of 21 knots (right). Both cases assume resilient mounting of engines.**

