# Peer review of "Modelling of ships as a source of underwater noise"

_Ocean Science, 2018_

## Referee Comment (RC1) · D. Wittekind (Referee) · 8 May 2018

The paper is well written and the procedures used and the result clear. I have the following comments The Wittekind model is valid for single screw ships only. Twin screw ships have in general lower propeller loading and a more homogenous wake field and therefore higher CIS. That is likely the reason that cruise liners and other twin screw ships appear quieter. Cruise vessels have CIS well above the mentioned 14 knots where diesel engine noise clearly prevails such that even if the propellers did cavitate they would be masked by diesel noise. The Wittekind model only considers 4-stroke engines be it for propulsion or as auxiliary diesels. 2-stroke engines are observed to have similar under water levels as resiliently mounted 4-stroke engines. 2-stroke engines may prevail at top speed but then they are masked by cavitation themselves.

If a heavy 2-stroke engine is taken as rigid mounted but with the same power-weightnoise relationship, diesel engine noise would be grossly overestimated. The model also does not cover CPP ships at low speed but these ships usually do not operate far away from their service speed in open water but in restricted areas they do. This may cause a very significant contribution in harbor approaches. I do not think that the above remarks if entered into the map would change very much but I recommend checking what the allegedly overestimated contribution of the 2-storke engines may do to it. Further remarks:  $\hat{a}$ Åć It would be interesting to know what the source depth was assumed to be  $\hat{a}$ Åć Maybe the Gigajoules could be converted into something more for feeling like the average equivalent URN level re 1  $\mu$ Pa in 100 m distance in 40 m water or something like this. It can be calculated by the educated but it would add to feeling what these numbers mean while reading  $\hat{a}$ Åć Could the authors add a graph showing the output (source level) of the Wittekind model for 2 or three typical ships?

---

## Referee Comment (RC2) · A. Farcas (Referee) · 29 May 2018

General comments

The manuscript presents a methodology for modelling the underwater noise source levels from shipping, with an example of application in the Baltic Sea for the year 2015. The topic is highly relevant in the context of ongoing efforts to monitor the ambient noise in the European and global waters, for the understanding of the anthropogenic contributions to the continuous underwater noise, of which the shipping noise is the main constituent. The proposed methodology combines a noise source model, incorporating several ship-specific parameters for predicting the source level of individual ships, with a ship traffic model (adapted from the authors' previous model for emissions

assessment) that essentially incorporates AIS data, in order to produce noise source maps. As the authors dutifully acknowledge, these source maps are not a representative description of the underwater noise, as they do not incorporate the propagation of noise. As such, they cannot be directly used to quantify the contribution of shipping noise sources in relation with the natural noise background. Nevertheless, they could represent a useful tool for a quick assessment of the pressure from shipping sources. While these source maps are not meant to used as direct input to propagation modelling, true shipping noise maps could be produced in principle by adding a sound propagation module to the model; as such, the methodology presented here is quite relevant for the mapping of the shipping noise itself.

Specific comments

It would be useful to compare or at least comment of the differences between the Wittekind source model used here and other models previously used in literature. For example, many shipping noise mapping methodologies might be based on the old Ross (1976) model that uses only the vessel speed and length to estimate the source levels of individual ships. Of course, a meaningful assessment would ultimately require a comparison of the noise maps based on the two source models and their statistical analysis; but even a comparison based on the noise source maps as produced by the methodology presented here might provide useful insights into the merits of using a more sophisticated source model.

The Block coefficient should be introduced or explained earlier in the text (currently it is explained that is a function of the hull shape only on its third mention, on page 5, line 26).

I am not an expert on ship source models, but it seems to me that the machinery noise source level would scale with the engine power rather than engine mass (of course, with appropriate scaling factors for different engine types). It appears that for two and four strokes engines, these scaling factors are as such that one can replace engine

power with mass and use just one scaling factor (namely the coefficient 15, in equation 4). But for turbine machinery, this no longer holds, as the authors indicate that there is no correlation between engine mass and power. However, the important question here is if a correlation does exist between the source levels and the engine power; if this is indeed the case, then an appropriate version of equation 4 should be used for such machinery, rather than plugging in the same mass dependency with the arbitrary factor of 0.001 ton/kW.

The finding related to containerships (that they are responsible for 25% of the noise energy) is quite interesting, but I'm not sure what is meant by them representing "about three percent of the ships in the Baltic Sea during 2015" – is this 3% of the total number of ships ever reported in 2015 in the area? Is the disproportionately high contribution of containerships to the total noise energy due to e.g. the greater number of "active" days per ship in this category than for ships in other categories (that might have been present or active only sporadically during the year), or perhaps this is due to more subtle factors related to source characteristics of containerships?

The noise source "map" concept is a modelling product that has both the spatial dimensions and the dimension of time. Figure 3 presents a spatial output, cumulated in time; the subsequent figures show information that was also cumulated in time. It would be perhaps informative to present some outputs that expose the time-dimension, be it locally or spatially averaged, for different ship types or for all – if such outputs showed anything interesting or insightful.

Minor technical corrections

Missing space "battlefield(warships)" on page 2 line 11

On page 4, line 5, "for which 10 000 tons should be used" – this is not really a recommendation, but a definition – use more decisive language, like "which is 10 000 tons".

Both "tons" and "tonnes" appear in the manuscript – is this correct? (tonnes are un-ambiguous, being a S.I. metric unit, while tons could be either "short" or "long" though this is probably the British "long ton" which is 1016 kg, used even in US in the naval context, and closer to the metric "tonne" than the U.S. "short ton", which is 907.2 kg)

Page 4, lines 21-21, "because all two stroke engines the cylinder arrangement is of in-line type" – does not read well. On page 8, line 26, "methodology how underwater noise[. . .]" – perhaps use "methodology describing how underwater noise[. . .]"

The Summary section could be tweaked – it sounds a bit too informal, the style is too "oral" like the conclusions of a presentation (e.g. "Our conclusions concerning this work are the following.", "It is evident that routine monitoring is required.")

—————————————————

---

## Referee Comment (RC3) · A. Farcas (Referee) · 1 Jun 2018

I agree that Gigajoules are not the most intuitive units for presenting the energy of the noise sources, but a conversion to source levels re 1 $\mu$Pa @ 100 m distance is still hard to make sense of, since the energy is cumulated over 1 year and either integrated over the 0.32 km2 cell (for the map of Figure 3) or cumulated for all the ships of a certain type (Figure 5).

For a single ship (of a certain category) perhaps if would be interesting to know some sort of average source levels (e.g. SL @1m per average containership, averaged over the full year, or only over the active period); in terms of a map, the average noise levels in the field would certainly be interesting, but obviously these are not straightforward

to calculate (the propagation from each cell out to, say, 100 km would need to be computed, etc.)

---

## Short Comment (SC1) · 1 Jun 2018

A map of the underwater noise energy emitted by AIS-tracked and identified ships in the Baltic Sea in the year of 2015 was obtained by using the Wittekind ship noise model for which the required parameters were either queried from technical databases or estimated.

This is an important step towards quantifying one of the dominant anthropogenic inputs of underwater noise into the marine environment of the Baltic Sea and a good attempt to deal with the unavailability of some of the ship parameters needed for the Wittekind model.

While the uncertainty of the derived underwater noise energy is qualitatively discussed,

the paper may benefit from a quantitative evaluation of the uncertainty via some type of error model that takes into account the various uncertainties that were qualitatively discussed in the manuscript. An illustration of the ship type distribution (e.g. manuscript only mentioned 3% were container ships) may add value to the paper as well.

Detailed Comments:

Page 2, line 4: omit 'the' before propeller cavitation Page 3, line 13: 'The Wittekind noise source model. . ..' (add 'The') Page 6 line 9: please mention chosen grid cell area A in method section. Page 7 , line 8: consider adding the names of the ports and islands on the map in figure 3. Page 7, line 10: Containerships by themselves represent about. . . Page7, 13/14: Please explain why ships transit in 2015 slower than normal Figure 1: horizontal axis may be rescaled up to <450 tons for better visibility Figure 2: horizontal axis may be rescaled up to <2500 tons for better visibility Figure 3: cannot pick out any yellow or red colors. Rescaling of colorbar may bring out better the smaller-scale differences in shipping noise between shipping lanes, which are currently all green or light blue. Would also suggest to make the labels of the colorbar aligned horizontally or have a vertical colorbar for better presentation and ease of use. Figure 4: consider integrating the 'Other' stack into the pie chart: slices shouldn't be too small as the 7% slices look big enough

---

## Referee Comment (RC5) · J. Hallander (Referee) · 7 Jun 2018

The paper presents a methodology for modeling underwater noise emission from ships based on AIS data. The authors combine the Wittekind noise source model with an assessment model for ship traffic emissions. The quantification of the underwater noise emission from ships is a highly relevant topic today. The paper is well written and the methods are clearly outlined.

Regarding the source model, the authors have made a good effort to estimate the data required by the source model that is missing in the AIS data and ship databases. It would be interesting to see a comment (/motivation) on the choice of source model. I.e. to point out the advantages with this choice compared to older models in the

literature. As both the authors and Dr. Wittekind points out in his comment, the model is mainly intended for large ocean going vessels with a single propeller. It would also be interesting to hear the authors view on this in relation to other recent studies trying to improve parametric models of ship source level for the purpose of mapping underwater noise emission from ships, e.g. [1, 2].

Regarding the simulation methodology, it would be interesting if the authors could put their work in relation to other similar attempts, e.g. [3, 4]. Especially since some of the co-authors, here are also co-authors of [4].

A specific comment about the discussion on noise emission from CP propellers, page 7, line 31: The reference (Li & Hallander, 2015), which is a popular text (without references) in SSPA customer magazine, is not the original source. I think the paper/report by Wittekind [6] is better as a general reference on this well known phenomena. How the noise first decreases and then increases again with decreased speed for a CP propeller is illustrated in [7]. A case study of the noise radiation from a ship with CP propeller at design speed compared to a typical reduced speed is presented in [5].

[1] C. Audoly, AQUO European Collaborative Project, deliverable R2.9 " Ship Underwater Radiated Noise Patterns", 2015. http://www.aquo.eu/downloads/AQUO_R2.9_Ship_URN_Patterns_V1.0.pdf

[2] C. Audoly, C. Rousset, "Parametric models of ship source level for use in an underwater noise footprint assessment tool", Conference Oceanoise 2015, Vilanova i la Geltrú, Spain, 11-15 May 2015

[3] Thomas Folegot, Mapping anthropogenic noise in European waters: examples from the AQUO and BIAS European projects, Conference Oceanoise 2015, Vilanova i la Geltrú, Spain, 11-15 May 2015

[4] T. Gaggero, I. Karasalo, M. Östberg, T. Folegot, L. Six, M. van der Schaar, M. André, E. Rizzuto, "Validation of a simulation tool for ship traffic noise", IEEE-MTS Oceans'15

Conference, Genoa, Italy, 19-21 May 2015

[5] Hallander, J., Karlsson, R. and Johansson, T., 2015, "Assessment of underwater radiated noise, cavitation and fuel efficiency for a chemical tanker", OCEANS'15, Genova, Italy.

[6] Wittekind, D., "Noise radiation of merchant ships", DW-ShipConsult, 10 July 2008.

[7] Beek, T. van. "Technology guidelines for efficient design and operation of ship propulsors", Propulsor technology, Wärtsila Propulsion Netherlands BV.

---

## Author Comment (AC1) · 4 Sep 2018

We thank the reviewers for their thorough assessment of our manuscript. The detailed response to each point can be found in a separate file included as supplement. This archive file also contains a revised version of the manuscript.

Please also note the supplement to this comment:
https://www.ocean-sci-discuss.net/os-2018-48/os-2018-48-AC1-supplement.zip

---

## Author Comment (AC3) · 4 Sep 2018

We thank the reviewers for their thorough assessment of our manuscript. The detailed responses are included as a separate file in the attached supplement. This archive file also contains the revised version of the manuscript.

Please also note the supplement to this comment:
https://www.ocean-sci-discuss.net/os-2018-48/os-2018-48-AC3-supplement.zip

---

## Author Response (AR1)

We thank the reviewers for their extensive comments. Below are our answers (in red). Modifications to the manuscript are indicated below. We hope these answers clarify the motivation and choices made in this work and hope that the manuscript can be published in Ocean Science.

It should be noted that an error was found in one of the formulas (part after Eq (6), describing the Pref) which has been corrected. A new model run was required because this change had an impact on all noise source maps and tabulated numbers. The revised manuscript with all other edits can be found within the supplement zip file.

**Reviewer 1 (Wittekind):**

The Wittekind model is valid for single screw ships only. Twin screw ships have in general lower propeller loading and a more homogenous wake field and therefore higher CIS. That is likely the reason that cruise liners and other twin screw ships appear quieter. Cruise vessels have CIS well above the mentioned 14 knots where diesel engine noise clearly prevails such that even if the propellers did cavitate they would be masked by diesel noise

We thank the referee for the comments. The Baltic Sea fleet mostly consists of vessels with a single propeller. About 10% of the fleet operating in the Baltic Sea during 2015 has more than one propeller. Usually RoRo, RoPax, Cruise and icebreaker vessels normally use multiple propellers. If passenger vessels were given a higher CIS value in the model, it would make their contribution to noise energy smaller than what is presented in this paper. Conclusions of most significant shipping noise sources of this paper would remain unaffected. We tested the Wittekind formulas for RoPax vessels and indeed, the machinery part contribution may exceed the low frequency cavitation.

The Wittekind model only considers 4-stroke engines be it for propulsion or as auxiliary diesels. 2-stroke engines are observed to have similar under water levels as resiliently mounted 4-stroke engines. If a heavy 2-stroke engine is taken as rigid mounted but with the same power-weightnoise relationship, diesel engine noise would be grossly overestimated. I do not think that the above remarks if entered into the map would change very much but I recommend checking what the allegedly overestimated contribution of the 2-storke engines may do to it.

About 82% of the vessels encountered in the Baltic Sea during the year 2015 were equipped with 4-stroke engines. We tested the impact of changing the engine mounting parameter to overall noise emissions of different kinds of vessels. A small cargo ship with single propeller and a 4-stroke main engine represents a case for which the noise source model was originally intended. As requested by the reviewer, in one of his later comments below calling for graphics for noise contributions, we added several images to Supplementary Material Section of the manuscript. The Supplementary Material Figure S7 presents the example of a small general cargo ship.

Supplementary Material Figs S8a and S8b illustrate the impact of flexible/rigid mounting, pointed out by the referee, for a feeder container vessel (1500 TEU). This vessel operates a single 2-stroke engine and has a FP propeller. Indeed, as the referee pointed out in his comment, a large difference exists in Source Levels because of engine mounting. Assuming rigid mounting makes machinery noise the dominating component of noise up to 6 kHz in this example. This specific vessel uses MAN S50MC-C series engine and according to the project guide of the engine, installation on epoxy or cast-iron chocks is required.

Figures S9a and S9b illustrate the impact of resilient/rigid mounting on source levels of a 150 000 DWT bulk carrier. The manufacturer of the engine (MAN 6S70MC) indicates that it is designed for rigid

installations on epoxy chocks. We acknowledge the comment that the original Wittekind noise source model was not specifically designed for vessels with large 2-stroke engines.

According to Rowen (2003), most engines are rigidly mounted. However, the Baltic Sea fleet significantly differs from the composition of the global fleet because of size restrictions of vessels. The handbook for diesel engines by Kuiken (2008) lists resilient mounting as the norm for category 1-2 engines, which are high- and medium speed diesels. According to Kuiken, engines in category 3 (medium to large 4-stroke engines) can be either resiliently or rigidly mounted, but majority of category 4 (large 2-stroke) engines are usually rigidly mounted. Indeed, technical manuals for Wartsila 32, 46 and 50 series, mentions that these can be installed with both options, but if resilient mounting is desired, the manufacturer needs to be consulted indicating that this is not necessarily the default option but rather an exception.

We have expanded the discussion of this issue in the manuscript to include the limitations of the original noise model (Section 3.3) and the justification for the assignment of engine mounting parameter. We also corrected a typo below Eq (4) concerning the assignment of the engine mounting parameter (Section 2.4).

It would be interesting to know what the source depth was assumed to be

This is irrelevant as the emitted power is independent of source depth.

Maybe the Gigajoules could be converted into something more for feeling like the average equivalent URN level re 1 \_Pa in 100 m distance in 40 m water or something like this. It can be calculated by the educated but it would add to feeling what these numbers mean while reading

One way to follow the reviewer is to add the following information: A source emitting 1 MJ during one year corresponds to a continuous monopole source with a SPL of approx. 156 dB re 1microPa at 1m, assuming that the free-field approximation is valid. The purpose of this paper is to report a methodology for noise source maps and energy emitted. Adding distance dependency to source maps would give a rough indication of noise propagation and affected areas. This was not the focus of this paper, however, because propagation modelling was done using the point source description for each individual ship in the area. Propagation studies will be published as a separate manuscript at a later stage.

We have added the description above to the end of Section 2.6 (Noise source maps)

Could the authors add a graph showing the output (source level) of the Wittekind model for 2 or three typical ships?

Below is a collection of noise source graphs for various kinds of ships. Figure 1 represents a general cargo vessel with a 4-stroke engine and a single propeller. Figures 2a-b and 3a-b respond to the previous question of the referee concerning the engine mounting parameter selection. Figure 4-5 contain two examples of RoPaxes with more than one propeller. One of the test cases (Figure 4) represents a vessel type which is equipped with two electric propulsion units and four diesel generators to indicate the extreme case of a multi-engine, two propeller case with diesel-electric propulsion.

We have added these five noise source cases (in seven images) to Supplementary material. These illustrate the form of the noise source curves as a function of frequency using the approach described in this paper as well as the impact of using rigid/resilient mounting for engines.

Figure 1 Source levels for a 15 000 DWT General Cargo vessel with 4-stroke engine and a FP propeller at design speed of 14.5 knots.

Figure 2 a and b. Noise source levels for a 1500 TEU Container feeder vessel with a 2-stroke engine and a FP propeller assuming flexible mounting (a, left) and rigid mounting (b, right). Vessel traveling at design speed of 19.8 knots